# Linear Method for Diagnosis of Inter-Turn Short Circuits in 3-Phase Induction Motors

**Yeong-Jin Goh [1]** and **On Kim [2],\***

1   Department of Electrical &Automation, Suncheon Jeil College, Suncheon 57997, Korea; jericho90@naver.com
2   Department of Technological, AP-tech Company Limited, Gwangyang 57714, Korea
\*   Correspondence: jnukimon@gmail.com; Tel.: +82-10-3646-1633

**Abstract:** When a turn-to-turn short fault occurs in an induction motor, it will be accompanied by vibration and heating, which will have adverse effects on the entire power system. Thus, turn-to-turn short fault diagnosis of the stator is required, and major accidents can be prevented if an inter-turn short circuit (ITSC), which is the early stage of a turn-to-turn short, can be detected. This study reinterprets Park's vector approach using Direct-Quadrature(D-Q) transformation for the linear separation of ITSCs and proposes an ITSC diagnosis method by defining the magnetic flux linkage pulsation and current change in the event of a turn-to-turn short. It is difficult to diagnose because the turn-to-turn short current change in an ITSC is considerably different from the induction motor loss. Hence, it was found through analysis that when the current change is considered through an analysis of the relationship between inductance and the winding number, the ITSC current becomes slightly smaller than the steady-state current. This was verified using the D-Q synchronous reference frame over time. We proposed a linear separation of the ITSC diagnosis from the steady state by considering the minimum values of the pulsating current as feature points.

**Keywords:** inter-turn short circuit (ITSC); induction motor; stator fault; turn-to-turn; D-Q synchronous reference frame

## 1. Introduction

Induction motors are widely used in many industries because they have a simple structure and are easily applicable to a system [1]. However, such motors are exposed to stress because they have to rotate periodically for long intervals. The motors are checked in accordance with a maintenance schedule to prevent faults, but sudden faults still cause considerable financial loss in industry.

The causes of induction motor faults can largely be classified into electrical and mechanical defects. Electrical defects are the main causes of stator and rotor faults, whereas mechanical defects are major causes of bearing faults. Mechanical defects account for 41% of all faults of induction motors, and electrical defects account for 45% of all faults, including 36% of stator faults and 9% of rotor faults [2,3].

In particular, stator faults are caused by dielectric breakdown, and the stator winding insulation is degraded by various causes such as coil movement, thermal stress, overburden, and mechanical vibration. This degradation of the winding insulation eventually leads to turn-to-turn short faults [4].

When a turn-to-turn short fault occurs, a large circulating current and high heat are generated in the fault circuit [5,6]. Most of these faults are caused by various stresses acting on the stator, such as thermal, mechanical, electrical, and environmental stresses [7].

Therefore, early diagnosis is urgently needed because the detection of inter-turn short circuits (ITSCs) can prevent sequential damage of the motor, minimize motor stop time, and the need for additional manpower and repair cost, thus avoiding considerable economic loss [8–10]. In the past,

detection methods using vibration and noise sensors were applied, but the sensors are expensive and additional circuits need to be designed and attached to the equipment.

Recently, non-invasive methods have been developed, e.g., acoustic, vibrational, thermal, magnetic methods [11–13]. Non-invasive methods are capable of diagnosing early faults without disassembly of the induction motor. Each type of signal has advantages and disadvantages. Measurements of acoustic and thermal signals are noninvasive. Acoustic and thermal signals can be measured without touching the motor. The method of selection of amplitudes of frequencies ratio 50 second frequency coefficient (MSAF-RATIO-50-SFC), MSAF-RATIO-50-SFC-EXPANDED, MSAF-RATIO-24-MULTIEXPANDED-FILTER-8, and shortened method of frequencies selection (SMoFS-15) are methods later used for diagnosis with the recent development of non-invasive sensor technology [14,15]. The disadvantage of the mentioned diagnostic signals is difficult processing.

Because of these inconveniences, the motor current signature analysis (MCSA), which is a sensorless method, is a recent focus of research [16–23].

The diagnosis of turn-to-turn shorts using MCSA has been researched in earnest using current spectra since 1989 [24,25]. Park's vector approach (PVA) was suggested in consideration of the fact that the induction motor must be diagnosed while it is rotating [25–27]. As PVA was developed in 1929, it became an analysis method that transformed three-phase orthogonal equations to two-phase orthogonal equations. In 1988, the readability of PVA was improved by enabling the diagnosis of faults by circular patterns [25,28,29]. However, fault diagnosis was still difficult due to the problems of iron loss and copper loss (of current) and because there was no clear difference between these and the ITSC condition. Consequently, research was conducted to define a distortion factor by extracting the maximum and minimum values during a short interval of operation [30–32]. However, the ITSC conditions were not much different from those at steady state. Hence, further research was conducted with a focus on the average shift of the distortion factor.

The extended Park's vector approach (EPVA) was researched to improve the visualization of PVA so that the PVA could be analyzed in the form of an ellipse by applying the square root [33–37]. However, the research was conducted in a complex way by adding algorithms to algorithms (e.g., applying it to the fast Fourier transform, FFT).

Existing studies have revealed that this is difficult because only the magnitude of the current is considered by reanalyzing the PVA to maintain its simplicity and readability. A method using change of the PVA phase angle was also researched [38–40], but it was difficult to directly solve the ITSC problem.

On the one hand, from the view point of the harmonics, EPVA analysis of the second time harmonic in the frequency in order to detect the current supply unbalance is suggested, but data by MCSA is not easy to interpret because the harmonics of the hi-order generates as a lot of information is merged when the 3 phase is converted to 2 phase; there is the difficulty that additional work must be performed to identify harmonics [33–37,41].

In addition, using the 3 phase source as the voltage total distortion (VTHD) of IEEE Std. 519, current unbalance factor (CUF) of IEC60034-1, voltage total harmonic distortion (VTHD), each voltage distortion harmonic (EV), current total harmonic distortion (CTHD), each current harmonic distortion (ECHD) characterized by a hybrid method in combination with vibration signature analysis, was studied. However, this method, which is the power system's point of view, is difficult for diagnosing ITSC and is highly dependent on vibration signature analysis [42–45].

Because the diagnosis of turn-to-turn shorts is important, a fault diagnosis method using the magnitude of a reverse phase component (impedance) was also studied. This was done by analysis of complex circuit modeling, even though the circuit was really complicated [46–54]. However, the algorithm became too complex due to the complex circuit and needed to be simplified. Machine learning is a method later used for diagnosis with the recent development of artificial intelligence (AI) technology [18,23,30,37,55–59]. AI-based methods are divided into feature extraction, fault identification, and fault severity evaluation. Compared with the existing general diagnostic methods,

an advantage of the artificial intelligence diagnostic method is that the diagnosis requires only minimal prior knowledge. There is no need to analyze the detailed model of the system or to model the fault. Most techniques use MCSA, vibration characteristics analysis, or a combination of both for fault detection. Many of these proposed techniques use FFT to analyze the spectrum in order to find the spectral characteristics of the fault. Once the feature extraction process to characterize the state of the motor is realized, the classification technology should be incorporated into the system. Neural network (NNs) classifiers have proved to be a good method in classification rate, accuracy and suitable hardware implementation. But the basic diagnosis method remains insufficient. Various recognition methods using AI technology such as k-nearest neighbors (k-NN), multi-layer perceptron (MLP), classification and regression tree (CART), and multi-class support vector machine (MCSVM) use the signal source of the voltage or current as it is, which makes it difficult to perform linear distinction of faults due to the noise of the power supply and environmental factors. Furthermore, 100% diagnosis is difficult because it is expressed as the failure rate of recognition. Moreover, applying the diagnostic apparatus is difficult because it shows differences that depend on the conditions and environments of industrial sites.

The challenge was to perfectly solve the turn-to-turn short fault problem of induction motors even if using a somewhat complex method because they can cause major accidents. However, improved readability and a simplified algorithm were required because online diagnosis must be possible. The difference between an ITSC and a turn-to-turn short needed to be examined by simplifying the change of magnetic flux for complex inductance in PVA terms.

The PVA was researched based on the synchronous reference frame of the D-Q transformation. This was drawn as a circle with the sum of the scalar values of the D and Q-axes. When this is checked over time, a pattern of direct current components can be obtained in an ideal steady state [60]. Furthermore, it can be estimated that the amplitude of the pulsation varies greatly depending on the degree of fault. However, the pulsation width of the ITSC is not much different from the actual steady state because of the iron loss and copper loss. Nevertheless, when all the information about the magnetic flux and current are moved to the D-axis of the D-Q synchronous reference frame over time, and only the zero component remains always on the Q-axis, analysis becomes easy.

Therefore, in this study, linear separation of the ITSC was performed, which was too difficult in previous studies. This was done by defining the relationship between turn-to-turn shorts and magnetic-flux linkage pulsation, as well as defining the change relation between the ITSC and the current.

## 2. Overview of Turn-to-Turn Shorts

In Figure 1, the short circuit in phase-A indicates the case in which a short circuit occurs between Turn 1 and a distant turn. The short circuit in phase-B indicates a short circuit between coils. The short circuit in phase-C indicates a fine short circuit between adjacent Turns 1 and 2.

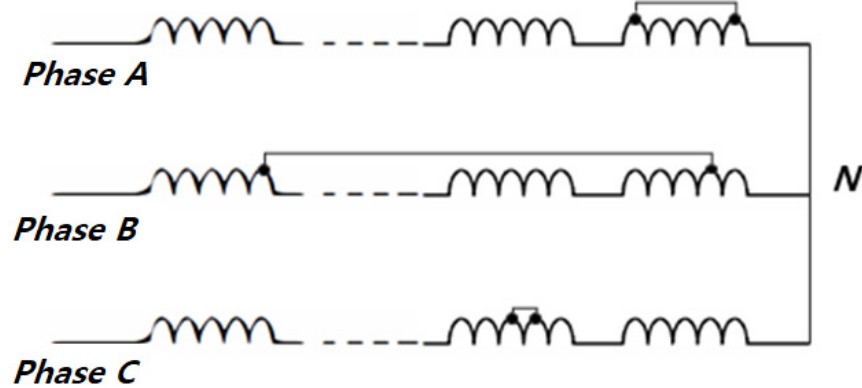

**Figure 1.** Short circuit in a slot.

These illustrate the two types of short circuits in a stator: coil-to-coil shorts and turn-to-turn shorts. When a fault like this occurs, an induction motor may generate severe vibration and heat. However, a fine turn-to-turn short is difficult to diagnose because there is no difference in the current between steady state and pulsation due to losses of the induction motor such as iron loss and copper loss.

Nevertheless, research on the detection of ITSCs is required because they can cause serious faults because of local heat generation and dielectric breakdown. Moreover, it must be possible to perform online diagnosis of 3-phase induction motors. This means that the motor is in rotation, and it is difficult to perform analysis in a time-varying state. Analysis can be facilitated using a D-Q transformation to change 3-phase signals to 2-phase.

## 3. D-Q Transformation

To change the variables of a 3-phase induction motor to variables on the orthogonal axes consisting of D, Q, and N axes is called coordinate transformation.

The D-axis is where the magnetic flux of the motor is typically generated; the Q-axis is symmetrical to the D-axis and is the reference axis. It is located before the D-axis in the rotating direction when the magnetic or other physical quantity of the induction motor rotates in the forward direction. Furthermore, the D-axis becomes the axis of the current that generates a torque. The N-axis is orthogonal to the D-axis and Q-axis in space and has a zero value in the motor.

In Figure 2a, when the D-Q orthogonal axes are fixed (do not rotate), the condition is called a stationary reference frame; in Figure 2b, when the axes rotate, it is called a rotating reference frame. The stationary reference frame is also called a stator reference frame. The rotating reference frame is a reference frame of which the axes rotate at the angular velocity of $\omega$, and its name and expression vary by the type of rotational angular velocity. For a representative example, a reference frame that rotates in synchronization with the rotor system is called synchronous reference frame, which is assumed with the PVA method.

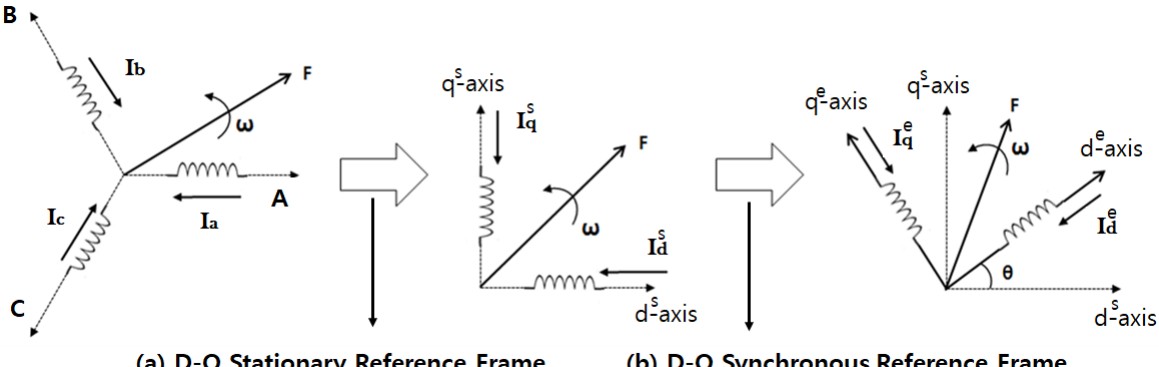

(a) D-Q Stationary Reference Frame     (b) D-Q Synchronous Reference Frame

**Figure 2.** D-Q transformation process: (**a**) D-Q stationary reference frame (**b**) D-Q synchronous reference frame.

In the process of transforming 3-phase to 2-phase, a synchronous reference frame can be obtained by the transformation of a stationary reference frame, as shown in Figure 2.

### 3.1. D-Q Stationary Reference Frame

In the stationary reference frame, the 3-phase variables $I_a$, $I_b$, and $I_c$ can be projected to the $d^s$, $q^s$ axis. The projected values of the variables to the $d^s$, $q^s$ axis can be obtained by Equations (1) and (2):

$$I_d^s = I_a \cos(0) + I_b \cos\left(-\frac{2}{3}\pi\right) + I_c \cos\left(-\frac{4}{3}\pi\right) \tag{1}$$

$$I_q^s = I_a \sin(0) + I_b \sin\left(-\frac{2}{3}\pi\right) + I_c \sin\left(-\frac{4}{3}\pi\right) \tag{2}$$

In Equation (1), $I_d^s$ is the current of the stationary reference frame projected to the $d^s$ axis and $I_q^s$ is the current of the stationary reference frame projected to the $q^s$ axis.

Because the $d^s - q^s$ axis is rotating, the 3-phase variable needs to be changed to a rotating $d^\omega - q^\omega$ axis variable. The general equation for transforming 3-phase data to orthogonal axes where the 3-phase variable is converted to a random angular velocity $\omega$ is as follows:

$$I_{dq} = T(\theta)I_{abc} \tag{3}$$

where $I_{dq} = \left[I_d^s, I_q^s\right]^T$, and $I_{abc} = [I_a, I_b, I_c]^T$. This is the result of generating the transposed matrix of angular velocity. The transformed $\mathrm{T}(\theta)$ can be expressed as follows:

$$\mathrm{T}(\theta) = \frac{2}{3}\begin{bmatrix} cos\theta & cos\left(\theta - \frac{2}{3}\pi\right) & cos\left(\theta - \frac{4}{3}\pi\right) \\ -sin\theta & -sin\left(\theta - \frac{2}{3}\pi\right) & -sin\left(\theta - \frac{4}{3}\pi\right) \end{bmatrix} \tag{4}$$

In Equation (4), $\theta$ denotes the rotation angle of the reference frame, and 2/3 is the coefficient of the transformation matrix. The effective values of voltage and current are the same, but the power and torque are reduced by 2/3. In this study, the coefficient of 2/3 was applied because the current data was used.

When Equation (4) is transformed into a matrix, we get the following equation:

$$\begin{bmatrix} I_d^s \\ I_q^s \end{bmatrix} = \frac{2}{3}\begin{bmatrix} 1 & -\frac{1}{2} & -\frac{1}{2} \\ 0 & \frac{\sqrt{3}}{2} & -\frac{\sqrt{3}}{2} \end{bmatrix}\begin{bmatrix} I_a \\ I_b \\ I_c \end{bmatrix} \tag{5}$$

The method of obtaining the stationary reference frame as in Equation (5) is also called Clark's Transformation.

This equation can be simplified as follows:

$$I_d^s = \frac{2I_a - I_b - I_c}{3} \tag{6}$$

$$I_q^s = \frac{1}{\sqrt{3}}(I_b - I_c) \tag{7}$$

Here, if the induction motor is ideal, the 3-phase winding is symmetrical. If the neutral point is not connected to another circuit, the sum of the physical quantities, such as the current of the 3-phase winding, is $I_a + I_b + I_c = 0$. Thus, we get $I_d^s = I_a$ from Equation (6). Therefore, the *d*-axis variable of the stationary reference frame is always the same as the phase A variable.

## 3.2. D-Q Synchronous Reference Frame

The rotating reference frame can be obtained from this stationary reference frame by applying the following equation with a random angular velocity:

$$\begin{bmatrix} I_d^e \\ I_q^e \end{bmatrix} = \begin{bmatrix} cos\theta & sin\theta \\ -sin\theta & cos\theta \end{bmatrix}\begin{bmatrix} I_d^s \\ I_q^s \end{bmatrix} \tag{8}$$

In Equation (8), $I_d^e$ is the *d*-axis current of the rotating reference frame and $I_q^e$ is the *q*-axis current. This transformation is called Park's Transformation.

The angle θ between the axis of the stationary reference frame and the axis of the rotating reference frame rotating at the angular velocity of ω changes over time, and can be obtained by integrating the rotational angular velocity of ω as follows:

$$\theta = \int_0^t \omega(\tau)d\tau + \theta(0) \tag{9}$$

where $\theta(0)$ is the initial angle at t = 0, and it is generally set as $\theta(0) = 0$. This synchronous reference frame rotates in synchronization with the rotor system.

Fault diagnosis using the method of identifying the distortion of the circle by expressing it as a Lissajous pattern (because the phase of $I_d^e$ and $I_q^e$ rotates together with the rotor system by a difference of 90 degrees) is called PVA. When this is observed as a movement over time, it is expressed as a DC value because the stator and rotor rotate in the same direction in the ideal case, and the size is the maximum value of the $I_d^e$ and $I_q^e$ AC values.

In a synchronous reference frame over time, the size on the *d*-axis is represented by the direction only and the components of the magnetic flux are moved to the *d*-axis. In this case, the *q*-axis always outputs a value of zero.

Figure 3 shows the changes in current caused by the D-Q transformation. Here, (a) represents the 3-phase input current and (b) represents the D-Q stationary reference frame changed from 3-phase to 2-phase. Furthermore, the result of applying the rotor velocity to Equation (9) is the D-Q synchronous reference frame over time in (c). As can be seen from (c), the rotor reference frame is expressed by the DC component in the ideal case.

If an imbalance occurs between two waveforms when the current waveforms of two phases $I_d$ and $I_q$ are expressed as a circular pattern, the circle is distorted.

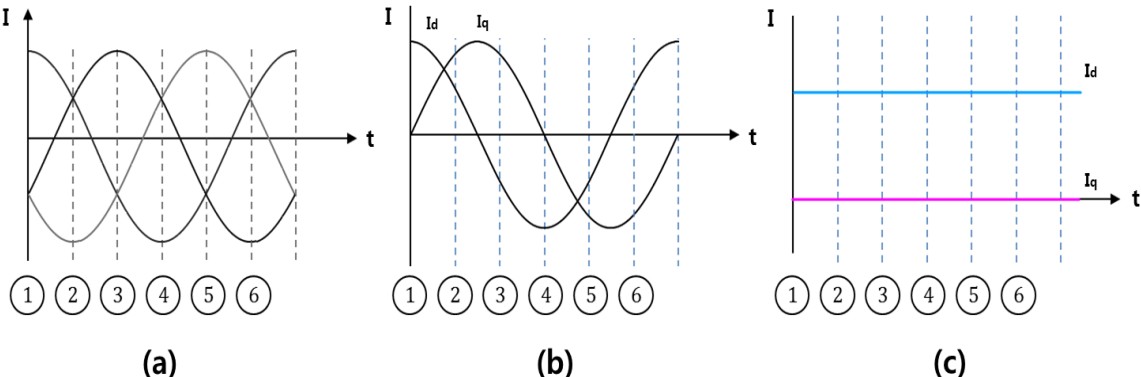

**Figure 3.** Changes to the current by the D-Q transformation: (**a**) 3-phase input current; (**b**) D-Q stationary reference frame; (**c**) D-Q synchronous reference frame over time.

On the other hand, when an imbalance occurs in the synchronous reference frame, which was synchronized with the rotor system by Equation (9), only incorrect changes in the flux or current can be extracted within the range of normal angular velocity ω. Furthermore, a pulsating phenomenon will occur for the DC component of $I_d^\omega$. Therefore, the D-Q synchronous reference frame over time was used for the diagnosis of ITSC in this study.

## 4. Turn-to-Turn Short and Inter Turn Short Circuit

### 4.1. Magnetic Flux Linkage Quantity according to a Turn-to-Turn Short

Industrial sites generally use 3-phase induction motors that are difficult to analyze when faults occur. Therefore, the initial 3-phase condition is usually reduced to 2-phase to simplify analysis using a D-Q transformation.

As shown in Figure 4a, in the event of a turn-to-turn short in one phase, an independent closed circuit to $R_s$ is generated, and another independent closed circuit occurs in the D-Q transformation, as shown in Figure 4b.

If the turn number at which the short circuit occurred is $N_S$, the winding turn number at which no short circuit occurred is $N_N$, the total number of turns is $N_T$, then $N_T = N_N + N_S$. The shorted winding and a non-shorted winding have the same inductance and direct interference occurs.

This interference causes an induced electromotive force in the shorted winding when magnetic flux exists with the same inductance, and this induced electromotive force generates the short circuit current $I_S$. When a turn-to-turn short occurs, $N_S$ is large, and the induced electromotive force is large as a result. Then the short circuit $R_S$ becomes small and a large short circuit current $I_S$ is generated. As shown in Figure 5a, due to the generation of $N_S$, two inductances exist in one phase, as shown in Figure 5b, and the magnetic fluxes link the two inductances.

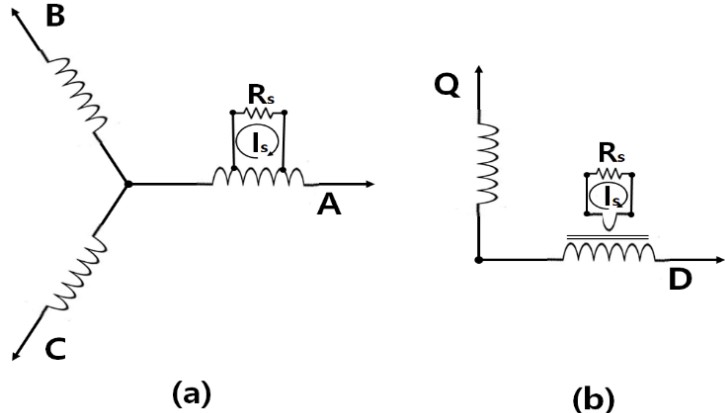

**(a)**　　　　　　　　　　　　　　　　**(b)**

**Figure 4.** (**a**) Occurrence of turn-to-turn short of phase R and (**b**) D-Q transformation equivalent model of a turn-to-turn short.

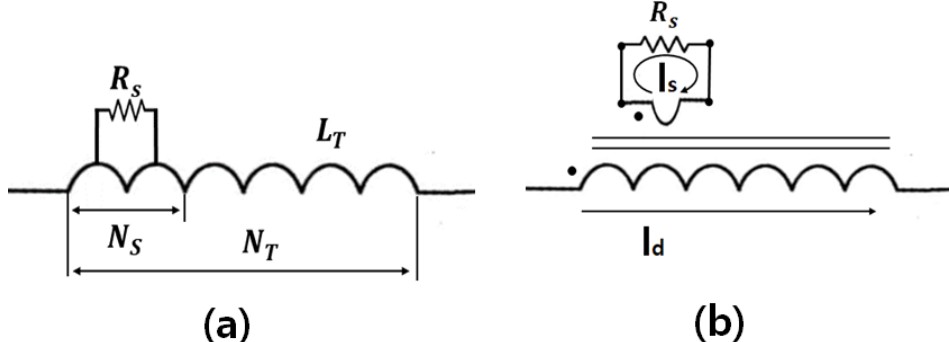

**(a)**　　　　　　　　　　　　　　　　**(b)**

**Figure 5.** (**a**) Inductance by turn-to-turn short and definition of winding number and (**b**) two inductances by a turn-to-turn short.

If the inductance of one phase is $L_T$ according to the winding number, the inductance $L_S$ by short circuit can be expressed by the short circuit ratio as follows:

$$L_S = L_T \frac{N_S}{N_T} \tag{10}$$

Figure 5 gives fragmentary indication that the inductance of one phase will decrease and the phase current will increase. However, change of the current in the event of a fault can be known only by investigating various kinds of information including the inductance, winding number, and magnetic

flux linkage quantity. In the event of a fault, using the conditions inductance, winding number, and flux, the magnetic flux linkage quantity can be defined as follows:

$$\lambda_{1S} = N_T(\phi_{1S} - \phi_{2S}) = N_T\left(\frac{L_T I_d}{N_T} - \frac{L_S I_S}{N_T}\right) = L_T I_d - L_S I_S \tag{11}$$

$$\lambda_{2S} = N_S(\phi_{2S} - \phi_{1S}) = N_S\left(\frac{L_S I_S}{N_T} - \frac{L_T I_d}{N_T}\right) = L_T \frac{N_S}{N_T} I_S - L_S I_d \tag{12}$$

where $\lambda_{1S}$ is the magnetic flux linkage quantity of one phase, $\lambda_{2S}$ is the magnetic flux linkage quantity of the fault inductance generated at the turn-to-turn short, $\phi_{1S}$ is the flux of one phase, $\phi_{2S}$ is the flux of the fault inductance, and $I_d$ is the normal current of one phase.

When a turn-to-turn short occurs, two fluxes appear that resemble a two-winding transformer, and the quantities of the two magnetic flux linkages generate a leakage flux. The total leakage fluxes of the short circuit $\lambda_{ST} = \lambda_{1S} + \lambda_{2S}$, can be expressed as follows:

$$\lambda_{ST} = (L_T - L_S)I_d - \left(L_S - L_S \frac{N_S}{N_T}\right)I_S = (L_T I_d - L_S I_S)\left(1 - \frac{N_S}{N_T}\right) \tag{13}$$

Based on Equation (13) only, the total flux decreases by the result of $(L_T I_d - L_S I_S)$, and the total leakage flux diminishes further according to the short circuit ratio. It can also be seen that the more severe the short circuit is, the lower the current becomes. Furthermore, as the short circuit becomes more severe, up to twice the previous leakage current will be generated.

The short circuit current $I_S$ is generally greater than the phase current $I_S$, and the amount of decrease also becomes greater, as the short circuit becomes more severe.

This reduction has direct proportionality with the current change in the rotor system; thus, the pulsation width of the current increases when it passes over the location of the short circuit.

### 4.2. Short Circuit Current According to ITSC

Unlike the turn-to-turn short explained above, an ITSC is a phenomenon of a short circuit between two adjacent turns and has a very small $N_S$. The $R_S$ maintains a very large value due to insulation resistance and shows exponential attenuation, decreasing rapidly over time. Attenuation indicates the occurrence of a turn-to-turn short. Thus, major accidents can be prevented if the ITSC can be detected before the exponential attenuation appears, and this requires further research.

In Equation (13), $1 - (N_S/N_T)$ can be considered to be equal to 1 because $N_S$ is very small. $I_S$ can also be ignored by insulation resistance. Hence, the same result as the magnetic flux linkage quantity in steady state, $\lambda_T = L_T I_d$, is obtained. In other words, in the case of an ITSC, the sizes of pulsation by losses such as iron loss and copper loss are similar in steady state.

However, the current $\lambda_{ITSC}$ according to the total flux of ITSC, $\lambda_{ITSC}$, can be derived as follows by Equation (14):

$$I_{ITSC} = \frac{\lambda_{ITSC}}{(L_T - L_{ITSC})} = \frac{(N_T - N_{ITSC}) \times (\phi_T - \phi_{ITSC})}{(L_T - L_{ITSC})} \tag{14}$$

where $I_{ITSC}$ is the phase current, $L_{ITSC}$ is the loss inductance, and $N_{ITSC}$ is the loss winding number. Because the total flux changes have no big difference, the change of flux according to the ITSC, $\phi_{ITSC}$, can be considered to be equal to zero. When an ITSC occurs, both $L_{ITSC}$ and $N_{ITSC}$ show decreasing tendencies. The relationship between these two elements is similar to a ring-shaped solenoid and thus can be determined by the following equation:

$$L_{ITSC} = \frac{N_{ITSC} \times \phi_T}{I_{ITSC}} = \frac{\mu S N_{ITSC}^2}{l} \tag{15}$$

where $\mu$ is the conductor dielectric constant, $S$ is the cross section area of the stator core, and $l$ is the average length of the coil. From this, we can see that $L_{ITSC} \propto N_{ITSC}^2$.

In other words, in Equation (14), even if the magnetic inductance of $L_T - L_{ITSC}$ decreases, $\lambda_{ITSC}$ decreases at a doubled rate. Therefore, the phase current $I_{ITSC}$ according to the ITSC shows a decreasing tendency in general. However, in the case of a turn-to-turn short, the phase current increases again as it is combined with the short circuit current $I_S$.

## 5. Experiment and Discussion

### 5.1. Experimental Conditions

In this experiment, a 3-phase induction motor with the specifications as in Table 1 was used.

**Table 1.** Motor specifications.

| Description | Value |
|---|---|
| Power | 0.75 [kW] (1 [HP]) |
| Input Voltage | 220 [V]/380 [V] |
| Full Load Current | 3.8 [A]/2.2 [A] |
| Supply Frequency | 60 [Hz] |
| Number of Pole | 4 |
| Number of Rotor Slot | 44 |
| Number of Rotor Slot | 36 |
| Full Load Torque | 0.43 [kg·m] |
| Rated Speed | 1690 [rpm] |

In addition, the experiment was performed under the following conditions to consider changes in the load conditions.

- Operation of an induction motor by using inverter;
- Adjustment of the motor operation speed by using dynamometer;
- The initial speed—the rated speed 1690 [rpm];
- operating time—turn short per 30 [s];
- Sampling rate and the number of samplings: 10,000 [S/s], 10,000 [s];
- The frequency of samplings: 1 [s].

During this time, two coils close to the turn number of the winding were randomly short-circuited.

The current was measured using an i5s AC current clamp by Fluke and data were collected using a USB-DAQ 9215A with BNC by National Instruments.

Figure 6 shows the structure of the total system.

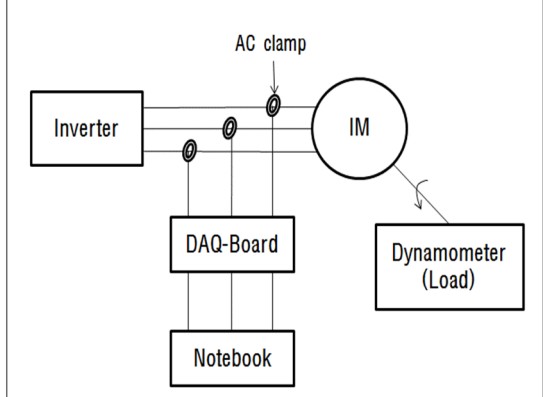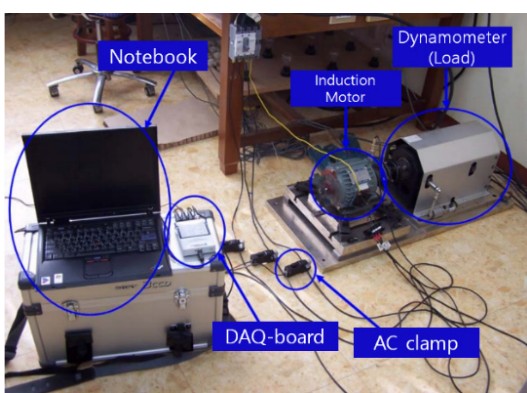

**Figure 6.** Experimental device configuration.

The measurement result of the input stage showed that the input power contained noise, as shown in Figure 7a. Thus, the cut-off frequency was set at 100 [Hz] by Butterworth third-order IIR filtering, and the filtering was processed as shown in Figure 7b.

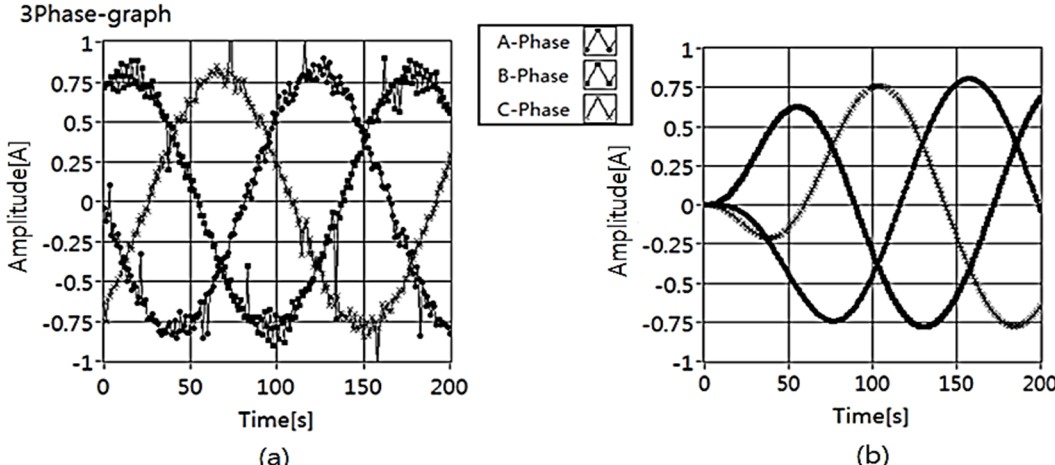

**Figure 7.** Filtering process for noise in the input power: (**a**) before filtering and (**b**) after filtering.

Furthermore, the short of the stator winding was configured artificially as shown in Figure 8a, and the winding was connected to an external tab as shown in Figure 8b.

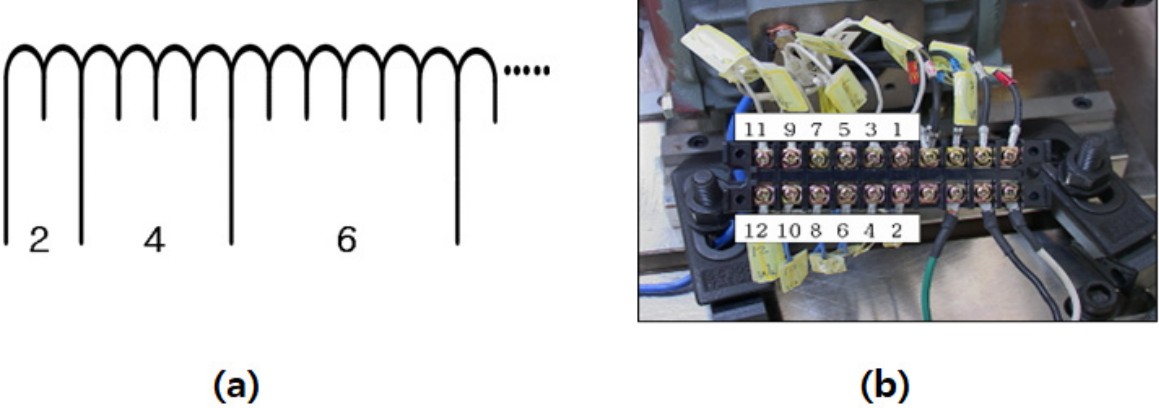

**Figure 8.** Method for configuring artificially turn short: (**a**) short of stator winding (**b**) external tab of the turn-short motor.

Table 2 shows a turn-to-turn short according to the tab connection. It shows the ITSC when turns no. 1–3 were connected. When turn no. 5 was connected, the turn-to-turn short increased to a 4-turn short.

**Table 2.** Turn short fault.

| No. | 1–3 | 3–5 | 5–7 | 7–9 | 9–11 |
|---|---|---|---|---|---|
| Turn short | 2(ITSC) | 4 | 6 | 8 | 12 |

### 5.2. Existing Studies-PVA

The PVA method represents the synchronous reference frame of D-Q transformation as a circular pattern and is considered an excellent method for online fault diagnosis. As shown in Figure 9, when the circle appears as an elliptical shape, it indicates the occurrence of a fault. Thus, the PVA method allows excellent visibility of a fault.

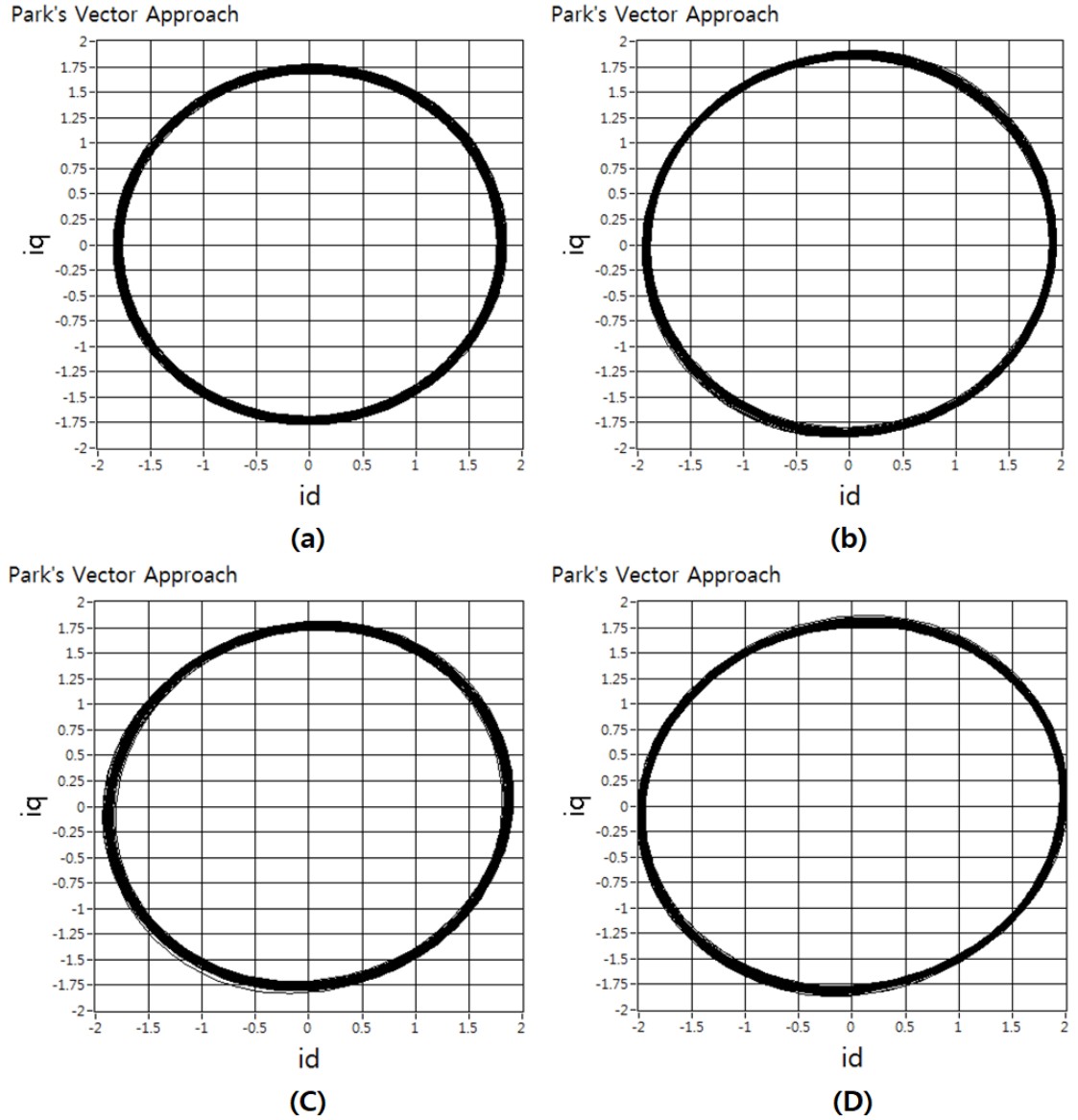

**Figure 9.** Park's vector approach (PVA) pattern: (**a**) steady state, (**b**) ITSC, (**c**) 4-turn short, and (**d**) 6-turn short.

To determine the degree of distortion of the circle, existing studies defined the distortion rate (DR) using the following equations:

$$r = \sqrt{I_d^2 + I_q^2} \tag{16}$$

$$DR = \frac{r_{max}}{r_{min}} \tag{17}$$

where r is the scalar value for the size of $I_d$ and $I_q$ of PVA, $r_{max}$ is the largest amplitude, and $r_{min}$ is the smallest amplitude.

Figure 10 shows the monitoring result of the fault diagnosis method using DR. It is the result of applying DR in Equation (17) per second.

As can be seen from Figure 10, linear separation of a 4-turn short and a 6-turn short is definitely possible when compared with the steady state, but ITSC is difficult to diagnose because it has no difference from the steady state.

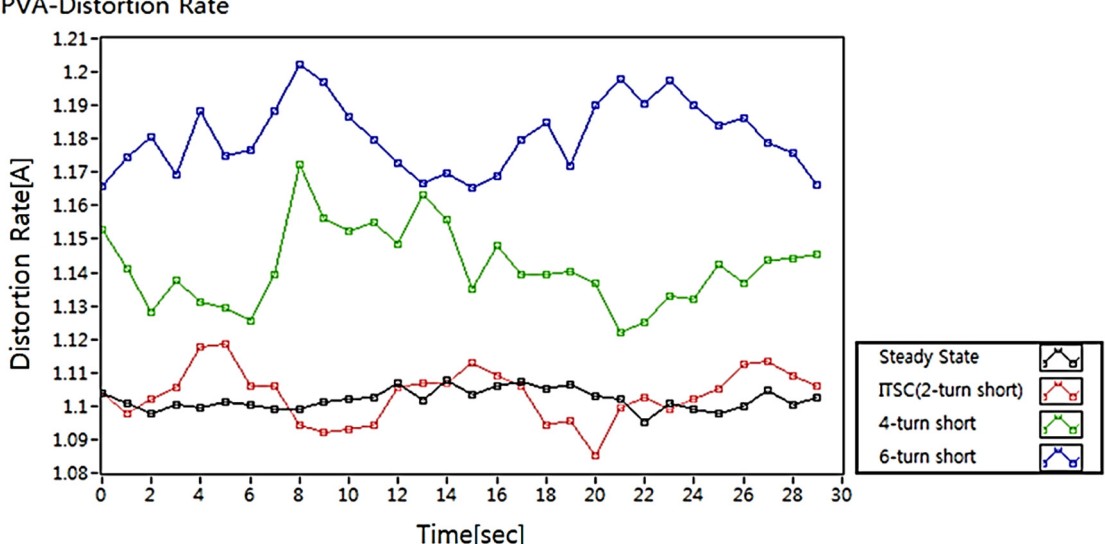

**Figure 10.** Stator fault diagnosis method using the distortion rate.

## 6. Proposed Method

It is difficult to diagnose ITSC using the PVA method because the maximum and minimum values based on the vector scalar sum of the *d*-axis and *q*-axis are expressed as ratios. This depends on the current pulsation and is based on the magnetic flux linkage quantity analysis described above.

This study indicates that the change of the short circuit current amount can be generated by linear diagnosis of ITSC, and a method using the D-Q synchronous reference frame over time was proposed to apply this procedure to experiments.

Figure 11 shows the monitoring results of a steady state and ITSC state using the proposed D-Q synchronous reference frame. It can be seen from this figure that the rotating induction motor in steady state contains an irregular AC component that pulsates due to loss. The pulsation in the ITSC state is not much different from the steady state; however, the overall current value decreases and the above-mentioned definition holds.

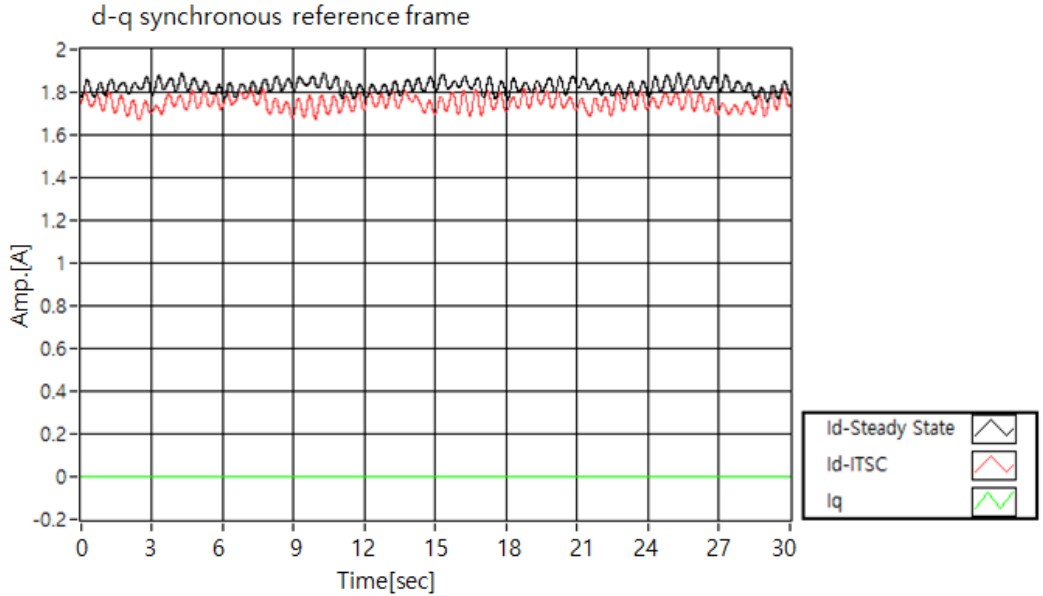

**Figure 11.** Inter-turn short circuit (ITSC) monitoring using a D-Q synchronous reference frame.

Figure 12 shows the results of measuring each turn-to-turn short for 30 s in the D-Q synchronous reference frame. Only the measured $I_d$-axis was monitored as the turn-to-turn short increased from steady state (black line), to ITSC (red line), to a 4-turn short (green line), and then to a 6-turn short (blue line). As can be seen in this figure, the ITSC current decreases, but the pulsation becomes severe from the 4-turn short. The maximum size of the $I_d$ current is similar to the steady-state current, but its minimum size is similar to that of the ITSC. When the turn-to-turn short increased further, the current became larger than the steady-state current range and severe pulsation was generated.

Figure 13 visually shows the changes by narrowing the range to 1 s for Figure 12. Table 3 shows the average values per second of the maximum, minimum, peak-to-peak, and average values measured in steady state and short circuit state from steady state to a 12-turn short.

**Table 3.** Changes in the maximum, minimum, peak-to-peak, and average current values according to the turn-to-turn short.

| No. | Max Value [A] | Min Value [A] | Peak-to-Peak [A] | Average Value [A] |
|---|---|---|---|---|
| Steady-State | 1.9026 | 1.7586 | 0.1439 | 1.8303 |
| ITSC | 1.8364 | 1.6717 | 0.1647 | 1.7503 |
| 4-Turn short | 1.8949 | 1.6531 | 0.2418 | 1.7722 |
| 6-Turn short | 2.0391 | 1.7363 | 0.3027 | 1.8949 |
| 8-Turn short | 2.2496 | 1.7564 | 0.4931 | 2.0141 |
| 12-Turn short | 2.3636 | 1.7514 | 0.6122 | 2.0715 |

As can be seen from peak to peak in Table 3, the peak to peak is 0.1439 A in steady state and 0.1647 A in ITSC. The ITSC is not much different from the steady state: the difference between the steady state and ITSC is approximately 0.02 A. However, in the results of the 4-turn short, the peak to peak is 0.2418 A, showing a difference of 0.1 A, and the amount of current increases by approximately 4.7 times relative to that of the ITSC.

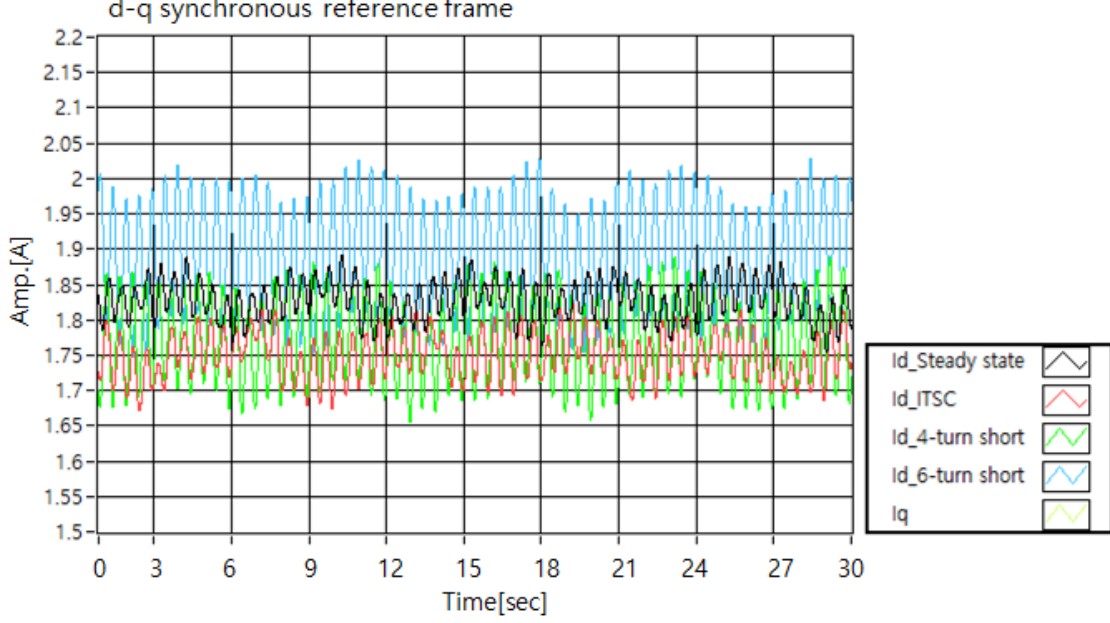

**Figure 12.** Comparison of turn-to-turn short sizes using the D-Q synchronous reference frame.

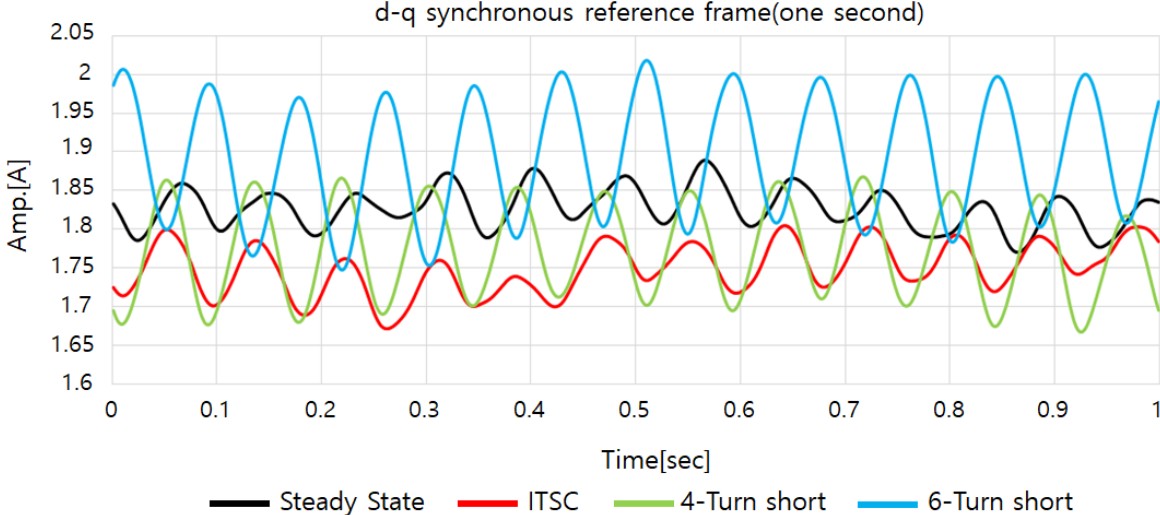

**Figure 13.** Comparison of the turn-to-turn short sizes using the D-Q synchronous reference frame (change rate per second).

The peak to peak represents the difference between the maximum and minimum per second. The existing method using the DR of PVA made it difficult to diagnose ITSC because it is based on this gap.

Figure 14 shows a graphic representation of the results in Table 3 for comparison.

The maximum current in Figure 14 shows a slightly decreasing trend in the ITSC, but increases sharply when the turn-to-turn short increases. However, the minimum value of the pulsating current increases very little even if the turn-to-turn short increases.

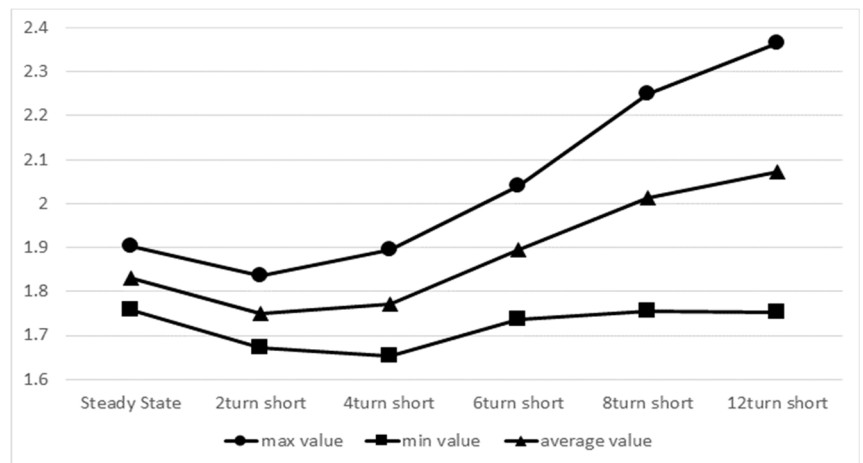

**Figure 14.** Changes in the maximum, minimum, and average current by short-circuit type.

Based on these results, the two-dimensional coordinates with pulsating maximum and minimum values as feature points are illustrated in Figure 15.

In Figure 15, the small circles indicate a steady state, the small squares indicate ITSC, the x marks indicate a 4-turn short, and the + marks indicate a 6-turn short. The maximum and minimum current values of the ITSC generally become smaller with similar pulsation sizes.

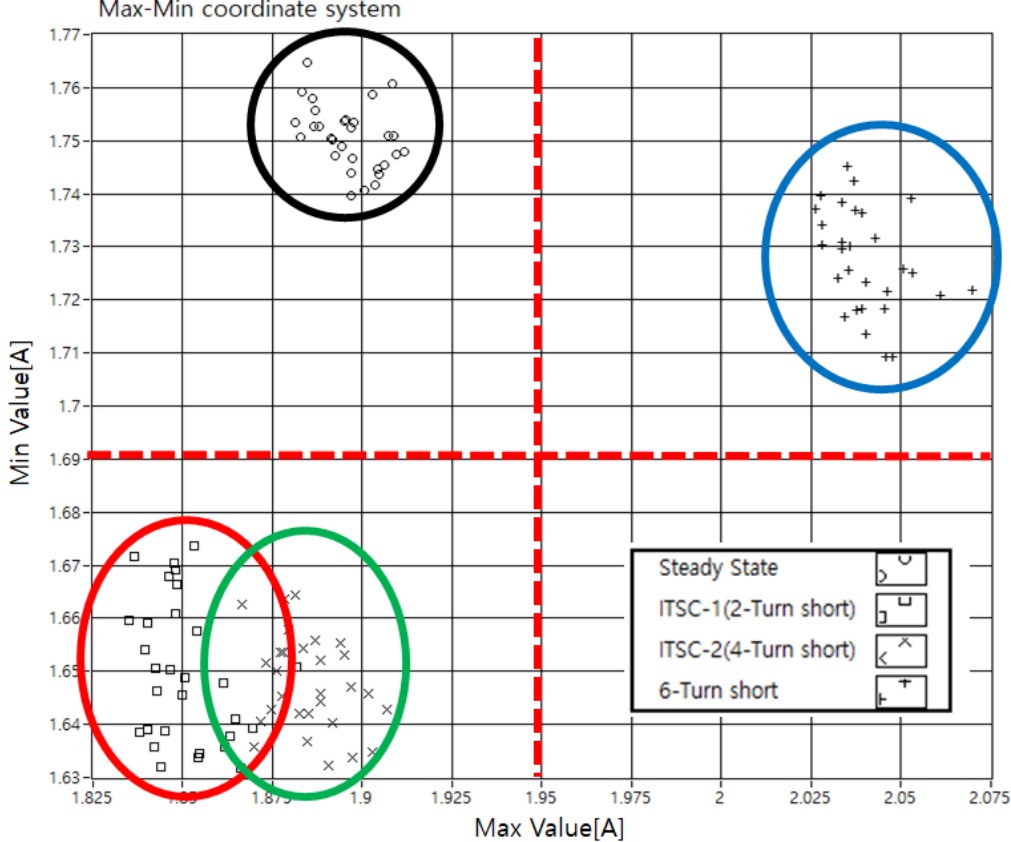

**Figure 15.** Max–min reference frame according to the turn-to-turn short.

When viewed based on the max value axis in the picture, the ITSC and the 4-turn paragraph occurs and the maximum change in the $I_d$ of the D-Q synchronous coordinate system can be seen through the area of the field and the red circle that is reduced finely, that is, the difference between the max value and the D-Q synchronous coordinate system, but it can be seen that it is insufficient to diagnose ITSC and 4-turn paragraphs; in the 6-turn paragraph it can be seen that it is possible to detect the failure by a large increase.

However, when viewed on the min value axis for ITSC and 4-turn paragraphs, it can be seen that the possibility of detecting a fine turn short circuit is significantly lowered.

The block diagram is shown in Figure 16. As shown in Figure 16, it can be seen that if the minimum value is used as a feature point, linear separation of ITSC from the steady state is possible, and if the maximum value is used as the feature point, a stator winding short can be diagnosed using the increased turn-to-turn short.

In recent years, a system capable of diagnosis in the situation of complex failure and speed variables using AI has been studied, but there is a lack of features for the AI. The proposed method is believed to be a new feature vector.

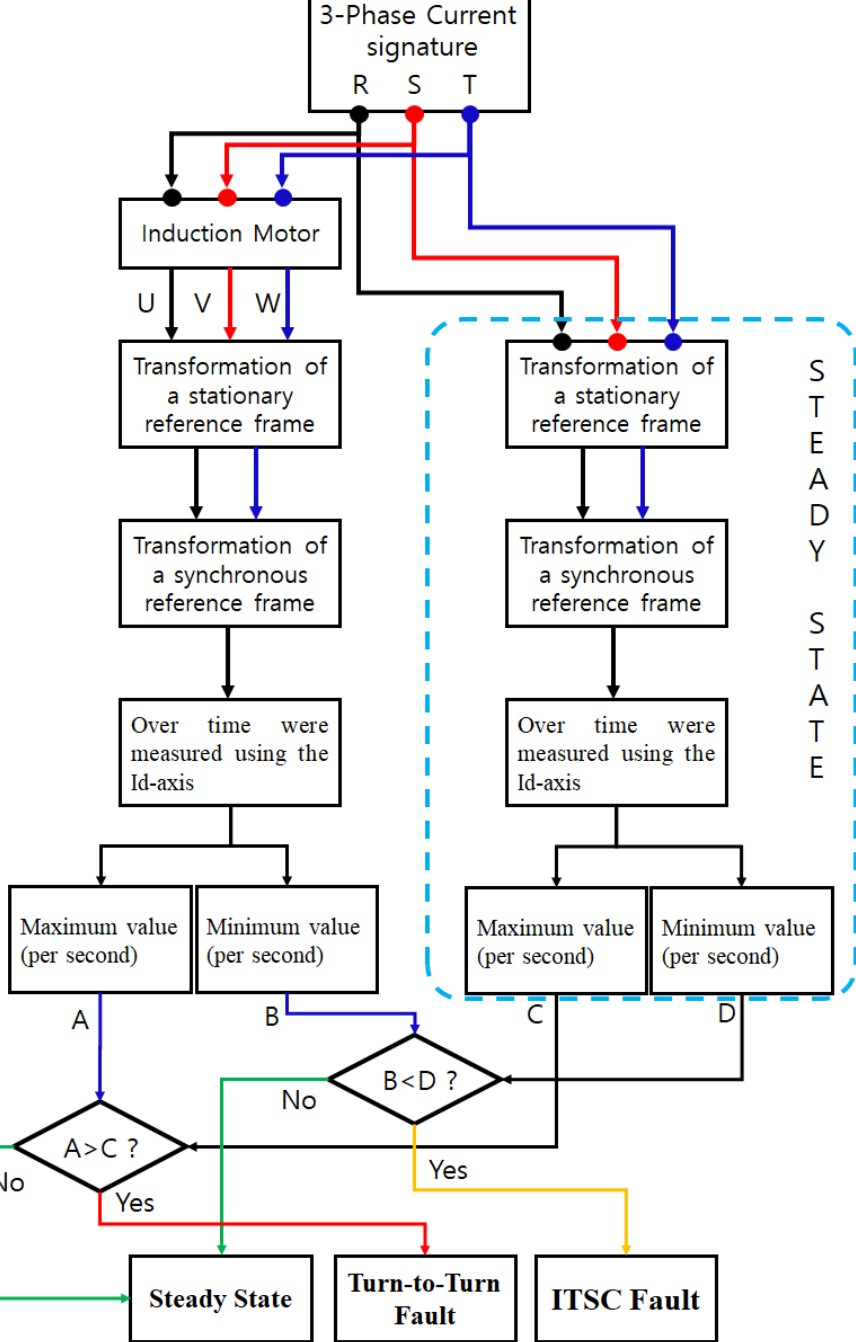

**Figure 16.** Block diagram for turn short fault in stator.

## 7. Conclusions

In this study, ITSC diagnosis and linear separation was proposed using the D-Q synchronous reference frame in a 3-phase induction motor to reveal shorts.

The existing PVA based on the D-Q synchronous reference frame draws a circular pattern, and a distortion of the circle results when a turn-to-turn short occurs but it is still difficult to diagnose ITSC because it is focused on revealing the distortion of the circular pattern while concentrating on the advantages of simplicity, good visibility, and excellent online diagnosis.

Therefore, we revealed that the existing PVA could not diagnose ITSC because it only used the gap information of the magnetic flux linkage pulsation. It was found through an analysis of the relationship between inductance and winding number that the current decreased slightly in ITSC.

To prove the interpreted results for ITSC, the maximum and minimum values over time were measured using the *Id*-axis of the D-Q synchronous reference. It was found that it was difficult to diagnose the ITSC due to the peak-to-peak pulsation gap. It was suggested that the minimum value must be considered for the diagnosis and linear separation of ITSC, and this was proven through experiments.

In the future, it is essential to optimize the manufacturing environment by applying smart factory applications that can predict potential accidents and prevent the recurrence of the same ones by quickly blocking and delivering various data to the upper-level system in the event of an unexpected fault.

**Author Contributions:** Conceptualization, Y.-J.G.; Data curation, O.K.; Formal analysis, Y.-J.G.; Funding acquisition, O.K.; Investigation, O.K.; Methodology, Y.-J.G.; Project administration, O.K.; Resources, O.K.; Software, Y.-J.G.; Supervision, O.K.; Validation, O.K.; Visualization, O.K.; Writing – original draft, Y.-J.G.; Writing – review & editing, O.K.

**Funding:** This research received no external funding.

**Conflicts of Interest:** The authors declare no conflict of interest.

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
