# Peer review of "Linear Method for Diagnosis of Inter-Turn Short Circuits in 3-Phase Induction Motors"

_applsci, doi:10.3390/app9224822_

Round 1

Reviewer 1 Report

Language: many sentences must be reformulated for clarification. Readers may need comparisons of this method with other established methods.  What is D-Q stationary Reference Frame (Figure 2 and 3) ?  Figure 2: why a-axis and b-axis are shifted in the figure? Notations for 3-phase systems should be the same in a paper. RST in Figure 1 and 4. ABC in Figure 2. UVW in Equation (1)-(7).  Figure 7 has no unit in vertical axis.  Figure 15: difficult to understand the graph. What are the dots in each group? Why 8-turn and 12 turn are not included?

Difficult to understand "To prove the analysis result, the maximum and minimum values over time were measured using the Id-axis" (in the conclusions). 

Conclusions are not clear and convincing. 

Author Response

Q1>What is D-Q stationary Reference Frame (Figure 2 and 3) ?

A1>To help explain, part 3 was divided into stationary reference frames and synchronous reference frames from line 150-172. We've also rewritten the lettering.

Q2>Figure 2: why a-axis and b-axis are shifted in the figure?

A2>a-axis modified as -axis, b-axis modified as -axis, we explained the reason at line 156.

Q3>Notations for 3-phase systems should be the same in a paper. RST in Figure 1 and 4. ABC in Figure 2. UVW in Equation (1)-(7).

A3>We are very sorry for our incorrect writing and we modified at line 118-219 .

Q4>Figure 7 has no unit in vertical axis.

A4>We are very sorry for our negligence and we modified at line 301.

Q5>Figure 15: difficult to understand the graph. What are the dots in each group?

A5>In a time-varying synchronous coordinate system, the maximum and minimum values are extracted one per second. We have marked the vertical axis as the maximum, and the vertical axis as the lowest 30 pieces of data (within 30 seconds). In general, in terms of current measurement and analysis (in terms of effectiveness), most measurements are made at maximum value. However, following the results, ITSC can be diagnosed as the minimum value, taking into account the synchronous coordinate calculation after time changed.

Q6>Why 8-turn and 12 turn are not included?

A6>In the actual experiment, we did the 6 turn-short experiment but the results showed severe shaking and fever, and the 8,12 turn-short experiment were even more severe. It doesn't make any sense.

Q7>Difficult to understand "To prove the analysis result, the maximum and minimum values over time were measured using the Id-axis" (in the conclusions).

A7>In order to easily understand, we modified "To prove the analysis" as "To prove the interpreted results for ITSC" In chapter 4, the pulsating changes and current changes in the magnetic field when ITSC occurs are defined, and the D-Q synchronous reference frame is used to confirm the result, and a maximum and minimum value is extracted per second from the D-Q synchronous reference frame.

Reviewer 2 Report

There is very few applications for directly connected induction motor, nowadays electrical machines are used mainly with variable speed drives, that produce additional harmonics. Effect of that harmonics are not discussed in the paper. In abstract (lines 9-11) you write: “When a turn-to-turn short fault occurs, it is accompanied by vibration and heat, which adversely affects the entire power system in severe cases.” Please provide additional description or reference for this fact. Please define abbreviations k-NN, MLP, CART, and MCSVM (line 70). It would be good to add additional overview for application of AI methods for turn-to-turn short fault diagnosis. Please explain using of definition D-Q Stationary Reference Frame. Conventionally it’s called αβ Stationary Reference Frame. With your symbols Figure 3 (b) and (c) are confusing In chapter 5 there is no information about load of the induction motor. It looks like no-load mode at Figure 7 (a) and full load at Figure 11. 1HP (0,75kW) motor is only a case study, I would recommend to verify proposed method with more higher power machine.

Suggestion from reviewer: put more attention on case study that you have provide. Make a proper analysis of turn-to-turn short fault diagnosis for low power induction machine (including literature review)

Author Response

Q1>There is very few applications for directly connected induction motor, nowadays electrical machines are used mainly with variable speed drives, that produce additional harmonics.

A1>It might be controversial, so we deleted it.

Q2>Effect of that harmonics are not discussed in the paper.

A2>we increased at lines 65-75.

Q3>In abstract (lines 9-11) you write: “When a turn-to-turn short fault occurs, it is accompanied by vibration and heat, which adversely affects the entire power system in severe cases.” Please provide additional description or reference for this fact.

A3>The abstract is supplemented

Q4>Please define abbreviations k-NN, MLP, CART, and MCSVM (line 70).

A4>we modified

Q5>It would be good to add additional overview for application of AI methods for turn-to-turn short fault diagnosis.

A5>We introduced overview from 81 to 90 lines

Q6>Please explain using of definition D-Q Stationary Reference Frame.

A6>It is divided into sections 3-1.

Reviewer 3 Report

Dear Authors, interesting article, but there are some corrections to make.
1) Equation 5: first row and third column, minus sign is missing
2) For the experimental part insert a table with all the parameters of motor. Also insert the information of the inverter (frequency of work, etc)
3) A table with the motor inductance and current values is missing after having short-circuited some coils for a check with the theoretical part.
4) Comment the experimental part better with the expected values.

Best Regards

Author Response

Q1) Equation 5: first row and third column, minus sign is missing

A1) misprint. Thanks.

Q2) For the experimental part insert a table with all the parameters of motor. Also insert the information of the inverter (frequency of work, etc)

A2) we added to section 5.1 as shown in Table 1

Q3) A table with the motor inductance and current values is missing after having short-circuited some coils for a check with the theoretical part.

A3) The inductance measuring device was not provided and the measurement was not carried out. We'll measure it after purchasing the equipment in the future.

Q4) Comment the experimental part better with the expected

A4) We added from 385 lines to 395 lines.

Round 2

Reviewer 2 Report

The authors have addressed my comments.

Author Response

Thank you for your kindness

Reviewer 3 Report

Dear Authors,
I thank you for correcting the paper. No other changes are required, it is accepted as presented.
Best Regards

Author Response

Thank you for your kindness.